# Phosphorylation controls RNA binding and transcription by the influenza virus polymerase

Anthony R. Dawson[1], Gary M. Wilson[2], Elyse C. Freiberger[3], Arindam Mondal[1¤], Joshua J. Coon[2,3], Andrew Mehle[1]*

**1** Department of Medical Microbiology & Immunology, University of Wisconsin–Madison, Madison, WI, United States of America, **2** Department of Chemistry, University of Wisconsin–Madison, Madison, WI, United States of America, **3** Department of Biomolecular Chemistry, University of Wisconsin–Madison, Madison, WI, United States of America

¤ Current address: School of Bioscience, Indian Institute of Technology Kharagpur
* amehle@wisc.edu

**Data Availability Statement:** All relevant data are within the manuscript and its Supporting Information files.

## Abstract

The influenza virus polymerase transcribes and replicates the viral genome. The proper timing and balance of polymerase activity is important for successful replication. Genome replication is controlled in part by phosphorylation of NP that regulates assembly of the replication machinery. However, it remains unclear whether phosphorylation directly regulated polymerase activity. Here we identified polymerase phosphosites that control its function. Mutating phosphosites in the catalytic subunit PB1 altered polymerase activity and virus replication. Biochemical analyses revealed phosphorylation events that disrupted global polymerase function by blocking the NTP entry channel or preventing RNA binding. We also identified a regulatory site that split polymerase function by specifically suppressing transcription. These experiments show that host kinases phospho-regulate viral RNA synthesis directly by modulating polymerase activity and indirectly by controlling assembly of replication machinery. Further, they suggest polymerase phosphorylation may bias replication versus transcription at discrete times or locations during the infectious cycle.

## Author summary

The influenza virus polymerase is a multifunctional enzyme directing viral gene expression and genome replication. Immediately following infection, the polymerase primarily performs transcription to make the viral mRNAs that program the replication cycle. The polymerase then shifts output to produce more copies of the viral genome at later stages of infection. The balance between transcription and replication is critical for successful infection. Here we identify phosphorylation sites within the viral polymerase and describe how these post-translational modifications control polymerase activity. Cellular kinases modify the viral polymerase. We identified a phosphorylation site in the catalytic subunit PB1 that selectively disables transcription, but not replication. We also describe a phosphorylation site in PB1 that disrupts binding to viral RNAs, disabling all activities of the

**Funding:** This study was funded by R01AI125271 (NIH NIAID) to AM and JJC, R35GM118110 (NIH NIGMS) to JJC, T32AI078985 (NIH NIAID) to ARD, and T32GM008349 (NIH NIGMS) to GMW. AM holds an Investigators in the Pathogenesis of Infectious Disease Award from the Burroughs Wellcome Fund. The funders had no role in study design, data collection and analysis, decision to publish, or preparation of the manuscript.

**Competing interests:** The authors have declared that no competing interests exist.

polymerase. These modifications may establish polymerases with specialized function, and help regulate the balance between transcription and replication throughout the viral life cycle.

## Introduction

All RNA viruses encode machinery both to express viral transcripts and to replicate genomes. Negative sense RNA viruses must first transcribe using virally-encoded RNA-dependent RNA polymerases (RdRPs) that are packaged into virions. The viral RdRP subsequently replicates the genome, often with the help of protein products from the recently produced mRNA. Regulating the balance and timing of transcription and replication is crucial for successful infection.

Viruses employ diverse strategies to control the abundance of virally-derived RNAs. Many RNA viruses rely on RdRP co-factors whose activity is dynamically regulated by post-translational modifications. For example, the Ebola virus polymerase is regulated by the viral transcription factor VP30. VP30 promotes transcription, whereas phosphorylation of VP30 results in its exclusion from transcription complexes favoring genome replication [1]. A similar strategy is employed by Marburg virus [2]. Dynamic phosphorylation of the M2-1 protein from respiratory syncytial virus regulates viral transcription. M2-1 is a transcriptional processivity factor whose function is proposed to require cycles of phosphorylation and dephosphorylation by cellular enzymes [3,4]. Phosphorylation also regulate global RNA synthesis. The phosphoprotein (P) from vesicular stomatitis virus is a polymerase co-factor. Phosphorylation on the N-terminus of P is important for transcription and replication [5,6]. The dynamic and fully reversible nature of phosphorylation enables localized and temporal control of viral proteins and may help progression through the infectious cycle. Phosphorylation of polymerase co-factors is thus a common strategy to regulate transcription and replication. However, influenza A virus and other members of *Orthomyxoviridae* do not encode polymerase co-factors and it remains unclear how their polymerases are regulated.

Influenza A virus contains eight negative-sense RNA genome segments packaged into virions as ribonucleoprotein (RNP) complexes. RNPs are double helical flexible rod-like structures containing the viral genome coated by oligomeric nucleoprotein (NP) and bound at both ends by the viral polymerase [7–10]. The viral polymerase is a heterotrimeric complex composed of the PB1, PB2 and PA subunits. Immediately following uncoating, RNPs are trafficked to the nucleus where synthesis of all virally-derived RNA occurs [11,12]. Infection initiates with a pioneering round of transcription from the incoming RNPs. The viral polymerase performs cap-snatching where a short capped oligonucleotide derived from the host is used to prime transcription [13,14]. Unlike transcription, replication initiates in a primer-independent fashion to create a positive-sense intermediate (cRNA) that serves as a template for vRNA production [15]. Replication requires concomitant assembly into newly formed RNPs to stabilize the viral genome [16]. Newly formed vRNPs can either be packaged into virions or serve as templates for additional rounds of transcription or replication.

The processes regulating polymerase activity are not fully defined. Some regulation simply requires the production of specific viral proteins or RNAs. The stable products from incoming RNPs are viral mRNAs, even though incoming RNPs are capable of making cRNA as well. Replication occurs later when newly synthesized NP is able to coat cRNA genomes and protect them from degradation [16]. Newly synthesized viral polymerase binds nascent RNA products and dimerizes with cRNP-bound polymerases to stimulate production of full-length vRNA

[17–21]. The nuclear export protein (NEP) and small viral RNAs (svRNA), both of which are made at later stages of infection, further bias the polymerase to replication [22,23]. Infection also induces broad changes in signaling cascades, and multiple host and viral proteins are regulated by post-translational modifications during influenza virus infection [24,25]. Like many other RNA viruses, phosphorylation of viral proteins plays a key role in regulating the influenza virus replication machinery. We have previously shown that phosphorylation of NP regulates *de novo* RNP assembly [26,27]. The protein kinase C (PKC) family, and PKCδ in particular, phosphorylates NP at its homotypic interface to block NP oligomerization. This is proposed to create a pool of monomeric NP that is subsequently licensed for oligomerization by a cellular phosphatase, possibly CDC25B [28].

PKCδ localizes to multiple subcellular sites including the plasma membrane, cytoplasm, and nucleus, with localization impacted by its activation status [29]. PKCδ acquires activating phosphorylation marks during influenza virus infection [27,30]. Additionally, apoptosis, which is induced during viral infection, promotes cleavage of PKCδ, leading to an abundance of active, nuclear-localized PKCδ [31,32]. Thus, PKCδ is likely active in both the nucleus and cytoplasm of infected cells. PKCδ phospho-regulates NP oligomerization and by extension the ability of the polymerase to replicate the viral genome [27]. These studies also provided intriguing data suggesting that the polymerase may also be phosphorylated. Whether phosphorylation directly regulates polymerase activity is unclear. Here we extensively map phosphorylation sites on the polymerase subunit PB1 and characterize their function. PB1 is the structural and catalytic core of the enzyme, and we define PB1 phospho-sites that inhibit RNA synthesis by blocking global catalytic function or genomic RNA binding. We also identified a regulatory site that split the function of the polymerase; mimicking phosphorylation at PB1 S673 suppressed transcription without altering genome replication. Viruses encoding phospho-ablative mutants at these positions displayed altered replication kinetics, whereas phospho-mimetic mutants did not replicate. These data demonstrate that phosphorylation directly regulates viral polymerase activity and may provide a mechanism to bias populations of polymerase towards replication or transcription.

## Results

### Phosphorylation alters activity of influenza virus polymerase

We had previously shown that PKCδ regulates RNP assembly by modifying NP and preventing premature NP oligomerization [27]. This work also revealed slower migrating species of the PB2 polymerase subunit that raised the possibility that the polymerase itself was phosphorylated in the presence of PKCδ. To test this possibility, we assessed the migrations patterns of PB2 before or after phosphatase treatment. We sought to study direct effects on the viral polymerase, but this cannot be done in the context of an RNP as PKCδ regulates NP function. We eliminated this confounder by using a short vRNA template (vNP77) that does not require NP for replication or transcription, and thus decouples RNP assembly from RNA synthesis activities [33]. The viral polymerase was expressed in cells with constitutively active PKCδ and vNP77 and immuno-purified samples were analyzed by western blot (Fig 1A). Slower migrating species were detected for PB2, confirming our prior results. Treating samples with phosphatase collapsed these species into a single band migrating at the expected position for PB2, suggesting that PB2 is phosphorylated. A shorter exposure of the same gel confirmed equivalent loading of PB2.

We then asked whether PKCδ expression affects RNA synthesis activities of the polymerase. The viral polymerase and vNP77 were expressed in cells in the presence or absence of constitutively-active PKCδ. Primer extension analysis of RNA extracted from these samples quantified

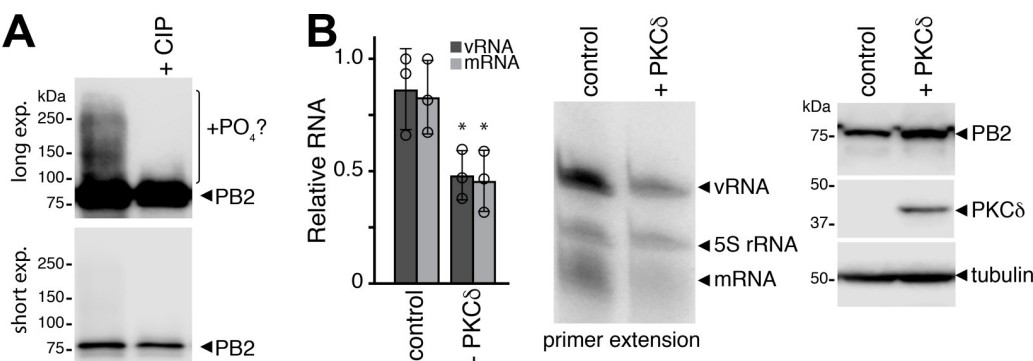

**Fig 1. PKCδ stimulates polymerase phosphorylation and inhibits polymerase activity. A.** The viral polymerase proteins and the mini-gene vNP77 were expressed in cells with a constitutively active form of PKCδ. The polymerase was immunopurified, mock treated or incubated with calf-intestinal phosphatase (CIP), and analyzed by western blot. Long and short exposures of the same blot are shown. A CIP-sensitive species that may indicate phosphorylated PB2 is indicated. **B.** Primer extension assays were performed on RNA extracted from cells expressing the viral polymerase, vNP77 and constitutively active PKCδ or an empty vector control. Viral replication (vRNA) and transcription (mRNA) were quantified, normalized to the 5S rRNA internal control, and presented relative to polymerase without PKCδ (mean of n = 3 ± sd; * = Student's t-test P < 0.05). Representative primer extension data are shown with western blots to confirm protein expression.

transcription (mRNA) and replication (vRNA) products. Co-expression of PKCδ significantly impaired production of viral transcripts and replication products, without altering viral protein levels (Fig 1B). Together, these data indicate that the viral polymerase is phosphorylated and these modifications may alter intrinsic polymerase activity.

## The polymerase core is phosphorylated at highly conserved sites

Given its potential to regulate polymerase activity, we performed a series of complementary experiments to extensively characterize polymerase phosphorylation (Fig 2A). We repeated experiments where phosphorylation was shown at affect polymerase function by expressing the viral polymerase, vNP77 and activated PKCδ in 293T cells. Polymerase was immuno-purified and analyzed by phospho-peptide mass spectrometry (MS) in two independent experiments. In parallel, we analyzed samples from infected cells. Polymerase phosphorylation status may vary across multiple rounds of infection; therefore, we collected samples from low and high MOI infections performed in A549 cells and analyzed whole-cell lysate. We also allowed for the possibility that phosphorylation patterns change throughout a single infection by analyzing RNP immunoprecipitations from both individual and pooled time points from synchronized infections. The amount of each sample used in the pooled lysate was adjusted to approximate similar levels of viral protein for all time points. These approaches allowed for high-confidence identification of phosphorylation sites on the viral polymerase (S1 Table).

We focused our analysis on PB1, the subunit that catalyzes RNA synthesis. A total of 13 phosphorylated residues were identified on PB1 in all experimental conditions, of which 8 were also detected in the context of infection (Fig 2A, S1 Table). Phosphorylation occurred primarily on threonine and serine residues, with only one phospho-tyrosine identified. Most of the phospho-sites were highly conserved in human H1N1, H3N2, and H5N1 strains (Fig 2A). A subset of sites or neighboring serine residues was also conserved in influenza B viruses and evolutionarily distant viruses circulating in bats. The potential impact of phosphorylation at each site was proposed based on polymerase structures (S1 Table). In addition to evaluating conservation of identified phosphorylated residues, we calculated the surface accessibility of each residue in multiple polymerase states: the monomeric polymerase bound to either vRNA

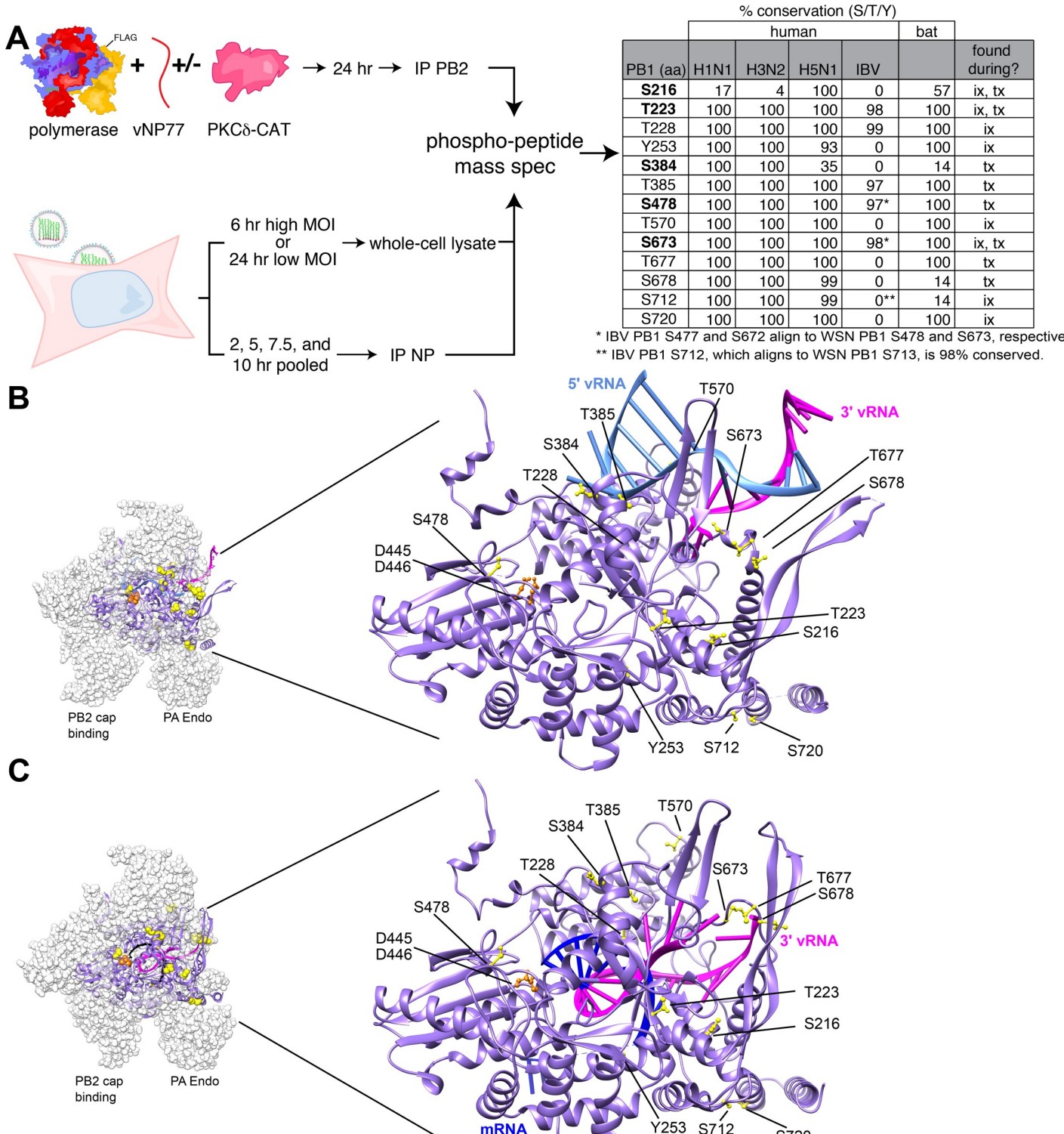

| | % conservation (S/T/Y) | | | | | |
|---|---|---|---|---|---|---|
| | human | | | | bat | |
| PB1 (aa) | H1N1 | H3N2 | H5N1 | IBV | | found during? |
| **S216** | 17 | 4 | 100 | 0 | 57 | ix, tx |
| **T223** | 100 | 100 | 100 | 98 | 100 | ix, tx |
| T228 | 100 | 100 | 100 | 99 | 100 | ix |
| Y253 | 100 | 100 | 93 | 0 | 100 | ix |
| **S384** | 100 | 100 | 35 | 0 | 14 | tx |
| T385 | 100 | 100 | 100 | 97 | 100 | tx |
| **S478** | 100 | 100 | 100 | 97* | 100 | tx |
| T570 | 100 | 100 | 100 | 0 | 100 | ix |
| **S673** | 100 | 100 | 100 | 98* | 100 | ix, tx |
| T677 | 100 | 100 | 100 | 0 | 100 | tx |
| S678 | 100 | 100 | 99 | 0 | 14 | tx |
| S712 | 100 | 100 | 99 | 0** | 14 | ix |
| S720 | 100 | 100 | 100 | 0 | 100 | ix |

\* IBV PB1 S477 and S672 align to WSN PB1 S478 and S673, respectively.
\*\* IBV PB1 S712, which aligns to WSN PB1 S713, is 98% conserved.

**Fig 2. The catalytic core of the viral polymerase is phosphorylated during infection. A.** Experimental design to detect phosphorylation of an active polymerase in transfected 293T cells or infected A549 cells. Samples were prepared as whole-cell lysate or immuno-purified proteins prior to phospho-peptide mass spectrometry. Most phospho-sites are conserved among circulating human influenza virus strains and highly pathogenic H5N1 viruses. PB1 sequences were aligned to WSN and the percentage where the indicated residue is a serine, threonine, or tyrosine is shown. Note that influenza B virus (IBV) residue numbering shifts relative to WSN. Whether phosphosites were detected during infection (ix), transfection (tx), or both is indicated. See S1 Table for all identified sites. **B-C.** Phospho-sites on PB1 surround the

catalytic core and template entry. The location of PB1 phospho-sites characterized in this study are modeled in yellow on (B) a pre-initiation complex (PDB 4WSB [44]) or (C) a transcriptional elongation complex (PDB 6T0V [36]). The motif C residues D445/D446 in the catalytic site are in orange, 5' vRNA is light blue, 3' vRNA is magenta, and transcription product is dark blue. Residue assignment and numbering based on WSN. PA and PB2 are shown as white in the space-fill representation, but are not shown in the close-up structure to increase clarity. Similarly, the 5' hook vRNA is not shown in C. See additional structures in S3 Fig.

[34] or cRNA [21], dimerized polymerase bound to cRNA [35], and polymerase during transcription elongation[36] (S2 Table). Residues corresponding to WSN PB1 T223, T385, S478 and T570 were inaccessible in all conformations analyzed, PB1 S384 was accessible in all conformations, and S673 was buried in the vRNA-bound structure but may be accessible in the monomeric. The remaining sites displayed an intermediate degree of accessibility. Highly-accessible residues may be dynamically modified, whereas the modification status of residues that are buried in a particular conformational state may be static. The polymerase samples multiple conformations depending upon its function, suggesting that accessibility and the modification potential of each residue will also vary depending upon polymerase function.

Some of the identified phosphorylation sites overlapped between our infected and transfected cells (S216, T223, S673), whereas others were identified only during infection (T228, Y253, T570, S712, S720) or when polymerase was co-expressed with PKCδ (S478, S384). We were most interested in sites that may directly affect polymerase activity, and not other functions of the polymerase like RNP export and trafficking. Thus, we focused on high confidence sites identified in polymerase activity assays where the primary function is polymerization and where we had shown that phosphorylation alters polymerase activity (Fig 1). We placed high priority on PB1 S216, T223, S384, S478 and S673. These phospho-sites could be broadly categorized into those that are proximal to the catalytic center (S216 and S478) or the template entry channel (T223, S384 and S673) (Fig 2B). All but S478 was identified in multiple experiments, and most sites were also identified during infection. Phosphorylation at PB1 T223 was identified during infection, confirming and extending the importance of prior work that had identified this phospho-site from transfected cells [37]. To our knowledge, none of the other phospho-sites have been previously reported.

## Phosphorylation status of PB1 impacts the ability of the polymerase to produce RNA and infectious virions

To assess the biological relevance of phosphorylation events at S216, T223, S384, S478 and S673, we attempted to rescue influenza virus encoding PB1 mutants harboring phospho-mimetic aspartic acid (D) or phospho-ablative alanine (A) mutations. All tested phospho-ablative PB1 constructs produced virus, whereas phospho-mimetic mutants at PB1 T223, S478 and S673 failed to yield virus despite multiple attempts. For PB1 mutants that support production of infectious virus, we measured viral gene expression during single-cycle infection of A549 cells (Fig 3A). Phospho-ablative mutants produced similar amounts of NP mRNA compared to WT, with the exception of PB1 T223A that had decreased transcription and PB1 S673A that exhibited a significant 5-fold increase in gene expression. Phospho-mimetic mutants displayed more subtle phenotypes, with PB1 S216D slightly above and PB1 S384D was slightly below WT levels. All rescued viruses were then assayed in a multi-cycle replication assay (Fig 3B). All viruses exhibited roughly similar replication kinetics at 12 and 24 hpi. However, PB1 T223A plateaued at peak titers ~10-fold lower than WT and the titers of PB1 S478A and S673A rapidly declined at 72 and 96 hpi to yield final titers ~10-fold lower than WT. PB1 harboring phospho-mimetics at position S216 and S384 yielded virus that replicates similar to WT virus despite producing disparate levels of NP transcripts (Fig 3A and 3B). The WT-level of replication for WSN PB1 S384D may be explained by the observation that the equivalent position is

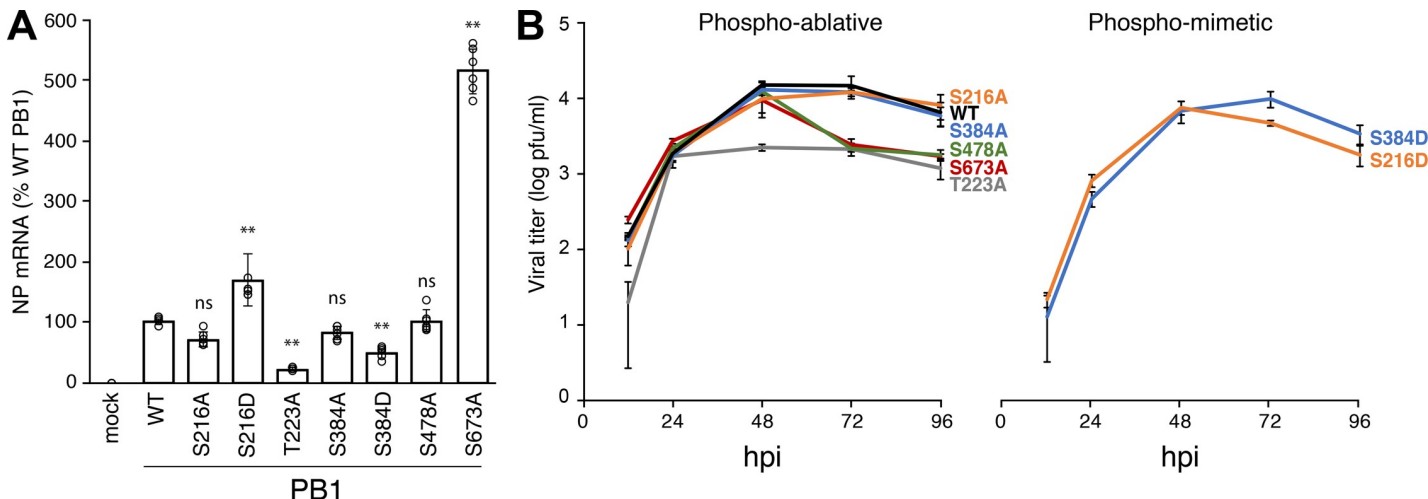

**Fig 3. PB1-mutant viruses identify phosphorylation sites that impact polymerase activity and viral replication. A.** PB1 phosphorylation both inhibits and enhances viral transcription. Single-cycle infections with PB1 phospho-mutant viruses were performed in A549 cells (MOI of 0.5 for 8h). RNA from infected cells was subject to qRT-PCR to detect NP and GAPDH mRNA. Fold changes ($\Delta\Delta C_T$) were determined in triplicate from 2 independent infections. (± SEM; ** = P <0.01 for one-way ANOVA with Dunnett's *post hoc* compared to WT). **B.** Multicycle replication kinetics of phospho-mutant viruses. A549 cells were infected at an MOI of 0.001. Viral titers were measured 12, 24, 48, 72, 96 hpi via plaque assay on MDCK cells. The WT reference curve applies to both the phospho-ablative and phospho-mimetic mutant viruses as all mutants were analyzed in the same experiment, but graphed separately to aid visualization (mean of n = 3 ± sd). P < 0.01 for one-way ANOVA at each time point. Statistics for ANOVA with Dunnett's *post hoc* pair-wise comparisons to WT are in S3 Table.

frequently aspartic or glutamic acid in polymerases from bat influenza A, human influenza B and C, and influenza D viruses. Constitutive phosphorylation at PB1 T223, S478, and S673 is incompatible with production of infectious virus, whereas the complete loss of phosphorylation at positions PB1 T223 and S673 also disrupts transcription and viral replication. These data suggest that differential phosphorylation of PB1 is important for successful infection.

Replication assays revealed PB1 phospho-residues important for the infectious cycle. We next focused solely on polymerase activity by performing primer extension assay for polymerases containing WT or phospho-mutant PB1 (Fig 4A, S1 Fig). WT polymerase produced significant amounts of viral mRNA, the replication intermediate cRNA, and vRNA indicating successful transcription and replication of the input genome. Phospho-mimetic mutants that failed to produce infectious virus also displayed defects in RNA synthesis. Polymerases with PB1 S223D or S478D exhibit profound defects with only background input vRNA levels and no detectable transcription or replication products. These results mirrored those obtained with a catalytically dead PB1 D445A/D446A mutant (PB1a) [16]. Remarkably, PB1 S673D replicated viral RNA, but showed a severe reduction in mRNA. Transcriptional defects for PB1 S673D were as strong as the previously described transcriptional mutant PB1 K669A/R670A (Fig 4A, S1 Fig)[38]. These data suggest that phosphorylation at PB1 S673 biases plus-sense RNA synthesis away from transcription and towards replication. PB1 S216D, S384D, and S673A generated transcription and replication products similar to WT PB1. Some of the differences in transcription detected during single-cycle infection (Fig 3A) were not fully recapitulated in primer extension assays (Fig 4A). This could be explained by the simplified nature of primer extension assays that lack viral factors that may modulate RNA production during infection [23]. Nonetheless, the combined defect in mRNA synthesis by PB1 S673D coupled with our inability to rescue virus encoding this mutant suggest a strong effect of phosphorylation at this position.

Polymerase activity requires multiple steps for successful replication and transcription, beginning with protein expression, trimer assembly, RNA binding, RNP assembly, and

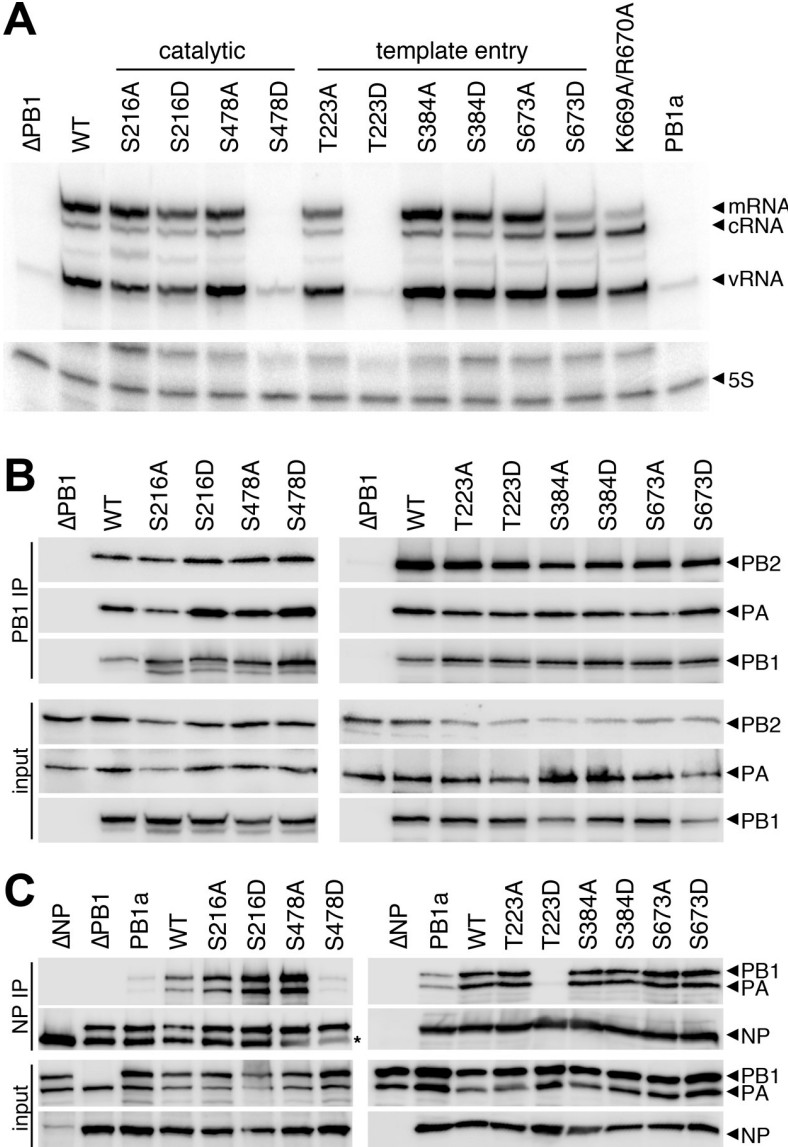

**Fig 4. PB1 phospho-mutants are defective in RNA synthesis and RNP formation. A.** Viral RNA synthesis was analyzed in primer extensions assays. RNA extracted from 293T cells expressing the viral polymerase, NP, and segment 6 vRNA was subject to primer extension analysis to detect transcription (mRNA) and replication (cRNA, vRNA) products. Primer extension of 5S rRNA was used as an internal loading control. PB1a, catalytically-dead PB1; PB1 K669A/R670A, transcription-deficient PB1. **B.** PB1 phospho-mutants form polymerase trimers. FLAG-tagged PB1, PB2, and PA were expressed in 293T cells and cell lysates were subject to PB1-FLAG immunoprecipitation. Immunoprecipitates and input samples were probed for PB1-FLAG, PB2, and PA. **C.** PB1 phospho-mimetics deficient in RNA synthesis fail to generate productive RNPs. NP immunoprecipitations were performed on 293T lysates generated as in (A). Immunoprecipitates and input samples were western blotted for PB1, PA, and NP.

ultimately the synthesis of new RNA products [39]. Primer extension reports on the cumulative success of this process. We therefore systematically investigated each step to identify regulatory points affected by PB1 phosphorylation. PB1 stability and polymerase trimer formation were assayed by expressing proteins in cells, immuno-purifying PB1, and probing for co-precipitating PB2 and PA (Fig 4B). All PB1 mutants expressed and formed trimers at approximately WT levels, independent of whether they were phospho-ablative or phospho-mimetic.

Thus, phosphorylation of PB1 at these sites does not control trimer assembly. Additionally, as polymerase trimers form in the cell nucleus, these data imply that defects in RNA production are not due to faulty nuclear import of polymerase subunits [40]. RNP assembly was next investigated by expressing RNP components in cells, immuno-precipitating NP and probing for co-precipitating polymerase (Fig 4C). Active polymerase will replicate the viral genome and amplify RNP assembly. We therefore utilized the catalytically dead PB1a to measure initial RNP formation that is independent of polymerase activity. PB1 mutants with defects in polymerase activity in primer extension assays failed to form productive RNPs, but the extent of the defect suggests different causes. PB1 T223D was completely excluded from RNPs, despite that fact that it forms trimers, suggesting phosphorylation at this position precludes incorporation into an RNP. PB1 S478D, however, assembled RNPs at low levels comparably to PB1a, indicating that RNP assembly *per se* is unaffected by this mutant. Rather, RNP assembly defects here stem from catalytic defects in the polymerase and not other steps in the process. The PB1 S673D phospho-mimetic does not alter RNP assembly, consistent with its ability to synthesize WT levels of genomic RNAs. In sum, loss of phosphorylation did not alter assembly and activity of RNPs in these assay. Conversely, mimicking constitutive phosphorylation at PB1 T223 or PB1 S478 prevented formation of productive RNPs and disrupted RNA synthesis. Finally, phosphorylation at PB1 S673 appears to toggle the viral polymerase primarily into replication mode.

## PB1 T223 phosphorylation inhibits vRNA binding and cRNA stabilization

Incoming RNPs synthesize both viral mRNA and cRNA, but cRNA is rapidly degraded; the polymerase and NP that would assemble into RNPs and protect cRNA from degradation have not yet been synthesized [16]. Whereas PB1 S478D was able to form low levels of RNPs, the complete failure of PB1 T223D was suggestive of defects in RNA binding. To test this possibility, we examined whether PB1 mutants could bind and stabilize cRNA during infection (Fig 5A). Polymerase with WT or mutant PB1 was expressed in cells prior to infection with WT virus. The oligomerization-deficient $NP_{E339A}$ was also pre-expressed to help stabilize cRNA while focusing the assay only on RNA made from the incoming RNPs. Cells were treated with actinomycin D during infection so only pre-expressed viral proteins were present. Primer extension showed that cRNA was stabilized by WT PB1 as expected [16]. Equivalent levels of vRNA in each condition confirmed efficient infection and delivery of vRNPs in all settings (S2 Fig). Trimers harboring PB1 S478D, which are unable to synthesize viral RNAs (Fig 4A), still stabilized cRNA yielding levels slightly higher than WT (Fig 5A). PB1 T223A also showed a minor increase in cRNA levels. However, polymerases with PB1 T223D exhibited a significant drop in cRNA stabilization. All of the other phospho-mutant polymerases stabilized cRNA to WT levels, consistent with their ability to replicate viral RNA. This was true even for PB1 S673D, which is replication competent but produces lower levels of mRNA (Fig 4A).

Viral RNA promoter binding is essential for stabilizing genomic RNA, suggesting that mimicking phosphorylation at PB1 T223 interferes with RNA binding. RNA immunoprecipitation assays were performed to measure promoter binding (Fig 5B). Polymerase and segment 6 vRNA were expressed in cells and polymerase was purified by immunoprecipitating PB1. Co-precipitating vRNA was detected by primer extension. vRNA co-purified with WT PB1, but not in its absence or when all polymerase subunits were excluded from the assay. PB1 T223D completely failed to bind vRNA, even though vRNA was readily detected for PB1 T223A. PB1 S478D bound vRNA, consistent with its ability to stabilize cRNA and form low levels of RNPs. If anything, PB1 S478D showed higher binding than both WT and PB1 S478A. These data parallel those from the stabilization assays. Multiple lines of investigation identify discrete defects caused by mimicking phosphorylation. Constitutive phosphorylation at PB1

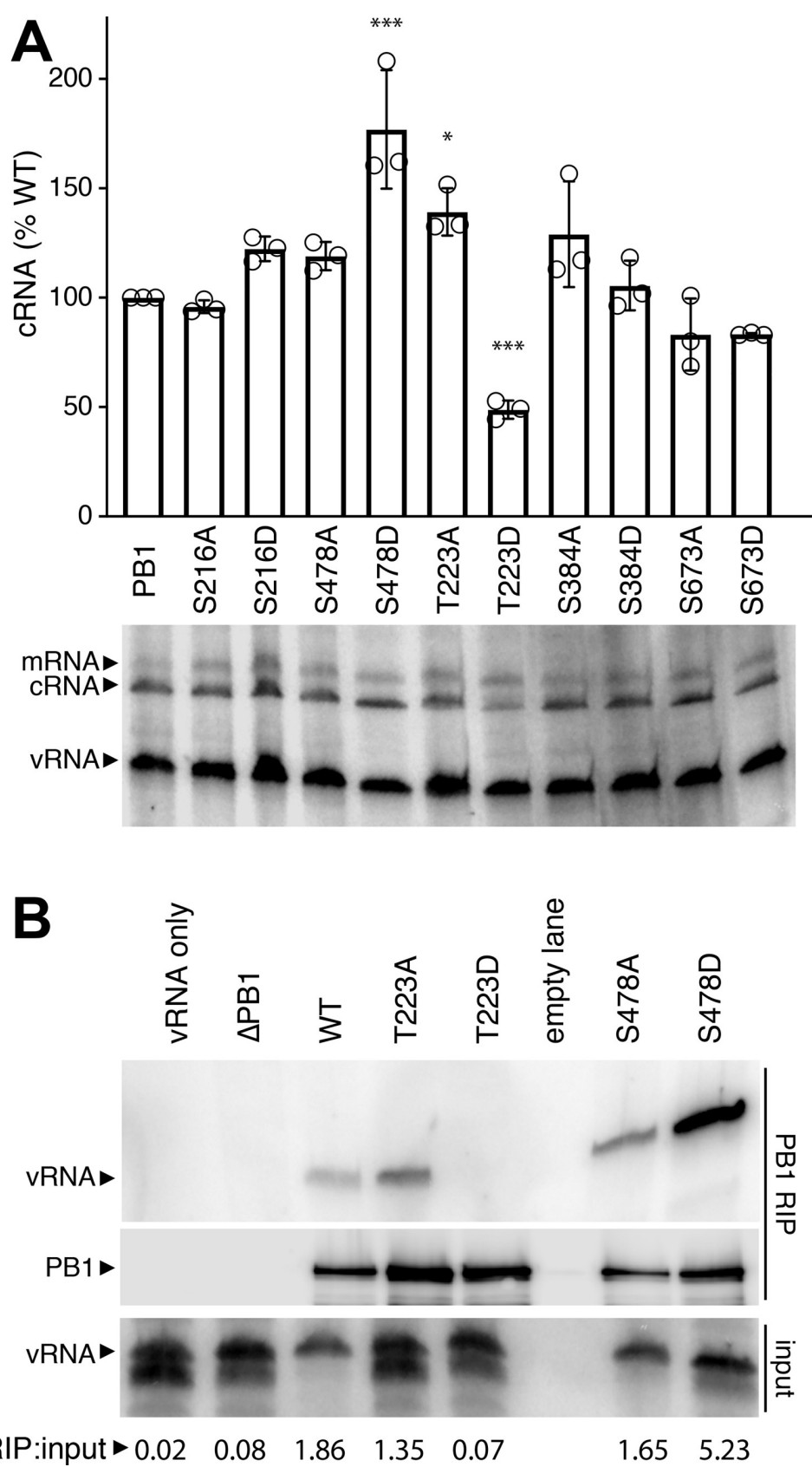

**Fig 5. Phosphorylation at T223 inhibits cRNA stabilization and vRNA binding. A.** PB1 phospho-mutants were tested in a cRNA stabilization assay. WT or mutant PB1 and oligomerization-deficient NP (NP$_{E339A}$) were expressed in 293T cells. Cells were treated with actinomyocin D (ActD) prior to infection, RNA extraction and primer extension analysis to detect transcription (mRNA) and replication (cRNA, vRNA) products. A representative primer extension gel is shown. cRNA levels were quantified, normalized to WT, and expressed as mean ± sd. * < 0.05, ** < 0.01, *** < 0.001 = P for one-way ANOVA with Dunnett's *post hoc* compared to WT. **B.** PB1 T223D fails to precipitate vRNA in an RNA-IP (RIP). PB1-FLAG, PB2, PA, and segment 6 vRNA were expressed in 293T cells. Cells were lysed and subject to FLAG immunoprecipitation. RNA extracted from immunoprecipitates and input samples was probed for the presence of segment 6 vRNA via primer extension analysis. Immunoprecipitated PB1 was confirmed via western blot. The ratio of radiolabel quantified for RIP and input is shown at the bottom.

S478 appears to disrupt catalysis without affecting RNA binding or initial RNP formation. Conversely, PB1 T223D was unable to stabilize cRNA or bind vRNA, suggesting that its inability to assemble RNPs and synthesize viral RNAs arises from defects in template binding.

## Discussion

Multiple mechanisms converge to regulate the influenza polymerase and bias production of either transcripts, the replication intermediate cRNA, or genomic vRNA. Here we reveal that phosphorylation of the polymerase directly regulates its activity. The core polymerase subunit PB1 was phosphorylated at multiple conserved sites, with clusters proximal to the catalytic center or the template entry channel. Mimicking phosphorylation at PB1 S478 or T223 disrupted global polymerase function by affecting catalytic activity or template binding, respectively. By contrast, phosphorylation at PB1 S673 preferentially suppressed transcription to create a replicase form of the polymerase. In all cases, mimicking phosphorylation at these key sites blocked the production of infectious virus, whereas ablating phosphorylation led to defects in replication. These data demonstrate that phosphorylation directly controls polymerase activity by either inhibiting or regulating RNA production.

The influenza polymerase performs diverse functions as transcription peaks early in infection, followed by production of the replication intermediate cRNA, and ultimately the final replication product vRNA [23]. The different functions are achieved in part by discrete conformations of the viral polymerase [39,41]. All of these require appropriate binding and positioning of the template [21,34,36,42]. Two of our PB1 phospho-mutants may affect this process. A phospho-mimetic at PB1 T223 completely disrupts RNA binding and stabilization of the nascent genome, ablating transcription and replication activity and the production of mutant virus (Figs 4 and 5). Our data confirm prior identification of this phospho-site and predictions that phosphorylation at this position might alter RNA binding [37,43].

Polymerase structures suggest that phosphorylation of S673 would also inhibit RNA binding [34,36,44]. Yet, phosphorylation at PB1 S673 appears to differentially suppress transcription without affecting replication (Fig 4). These data imply RNA binding remains intact, at least for replication. This conclusion is supported by similar phenotypes from the PA H510A polymerase mutant. PA H510 is apposed to PB1 S673 in the template binding and entry channel (S3A Fig). The PA H510A mutation selectively inhibits transcription, whereas replication is near WT levels [45]. That two mutants in this region both confer similar defects highlights its importance in properly positioning the template in the active site for transcription, and suggest that template binding or positioning constraints may be stricter for transcription than replication. More recent structures of the polymerase at later stages of transcription suggest another more direct mechanism for the transcription defects associated with PB1 S673D. While involved with template binding during transcription initiation, PB1 S673 remains located near the viral template during transcription elongation, polyadenylation, and pre-termination (S3B–S3D Fig). Transcription elongation causes the formation of the template exit

channel [36]. The PB1 thumb and PB2-N1 domains move and the PB1 priming loop is extruded from the active site to form a narrow channel with PB1 helix α20 on one side and portions of the priming loop and PB2-N1 on the other (S3 Fig). Importantly, PB1 residues 667–681 are reconfigured where they form new salt bridges, contribute to a three-strand sheet, and residues K669, R670, and R672 establish the basic floor of the channel (S3B Fig). PB1 K699, R670, and R672 have been shown to be specifically important for transcription (Fig 4 and [38]). Phosphorylation of PB1 S673 could conceivably prevent these conformational rearrangements, disrupt the basic nature of the channel, block the exit channel, or some combination of these. The cumulative outcome would be to preclude template exit. While this could explain the transcription defects associated with PB1 S673D, the fact that PB1 S673D is competent for replication creates an interesting contradiction. It raises the possibility that the template exit pathway during replication is more accommodating, or perhaps even fundamentally different, than the current structures of transcription complexes suggest. Additional work is needed to resolve this discrepancy with the potential to provide insight into key differences between transcription and replication cycles. Independent of the exact molecular mechanism, phosphorylation at PB1 S673 could be a way to establish replicase-specific polymerases.

PKCδ interacts with the viral polymerase and modifies NP to control RNP assembly [27]. Here we showed that phosphorylation was identified on PB1 S478 in cells expressing active PKCδ. Mutational analysis revealed that the phospho-mimetic PB1 S478D is functionally analogous to the well-characterized PB1a allele that mutates the conserved SDD motif in the active site: PB1 S478D retains the ability to bind the viral promoter, stabilizes cRNA, and forms initial RNPs, but is catalytically inactive (Figs 4 and 5). S478 lies in the NTP channel of the polymerase and phosphorylation would likely interfere with NTP transit or positioning [46]. This polymerase is catalytically inactive, but like PB1a may still impact overall polymerase output by functioning in *trans* to stimulate cRNP activity [19]. Conversely, RNPs harboring PB1 phosphorylated at S478 would be non-productive; thus phosphorylation may be an antiviral host response. It remains to be determined whether phosphorylation of PB1 S478 exerts antiviral activity or contributes to *trans* polymerase function during infection. Moreover, other phosphorylation events that we identified but did not study in detail here may also play multiple roles during infection (S1 Table).

Our studies reveal that constitutive phosphorylation largely inhibits specific polymerase functions, whereas phospho-ablative mutants are more tolerated. Phospho-ablative mutants at PB1 T223, S478 and S673 retained polymerase function, but exhibit dysregulated abundance and altered replication profiles (Fig 3). These data suggest that balanced phosphorylation at these positions is important for normal polymerase output. Moreover, we observed unusual replication curves for viruses encoding PB1 S478A and S673A, which exhibited WT kinetics through 48 h, but then declined rapidly at 72 and 96 hpi. However, PB1 S478A and S673A polymerases showed no defect in polymerase activity assays. The underlying cause for this altered replication is not known. The phospho-sites are all located at sites that are not immediately on the surface of the polymerase in the pre-initiation state (S2 Table). This suggests kinases modify PB1 during its translation or assembly into the trimer. The polymerase is highly flexible, and the surface accessibility of each residue varies according to the polymerase conformation. However, residues 223 and 478 remain buried in multiple polymerase conformations, raising the possibility that these modifications cannot be accessed by a phosphatase and are thus static. Instead of dynamically regulating the activity of an individual polymerase, phosphorylation at these sites might permanently assign a function and establish pools of specialized polymerases. Phosphorylation indirectly controls genome replication by regulating RNP assembly, and we now show that modifications on the viral polymerase directly control its activity to regulate product output.

## Materials and methods

### Cells, viruses, plasmids, and transfections

All experiments were conducted with A549 (CCL-185), HEK 293T (CRL-3216), MDCK (CCCL-34), or MDBK (CCL-22) cells acquired from ATCC. Cells were maintained in Dulbeco's modified Eagle's medium (DMEM; Mediatech 10-013-CV) with 10% FBS and grown at 37˚C in 5% $CO_2$. Cells were regularly verified as mycoplasma negative using MycoAlert (Lonza LT07-218).

All virus and virus-derived protein expression constructs are based on A/WSN/1933. Expression constructs for the viral polymerase and NP were described previously [47,48]. FLAG-tagged PB1 expression constructs was generated via restriction cloning to express PB1 with a C-terminal 3X FLAG tag. Mutations were introduced by inverse PCR and confirmed by sequencing. The catalytically dead PB1a (PB1 D445A/D446A) and transcription-defective PB1 K669A/R670A mutants were previously characterized [16,38]. Plasmid expressing the catalytic domain of PKCδ was previously described [49] (Addgene plasmid #16388).

Viruses were prepared using the pBD bi-directional reverse genetics system and pTM-All derivatives where multiple gene segments are consolidated on a single plasmid [48,50]. Rescued viruses were amplified on MDBK cells and titered on MDCK cells by plaque assay. When preparing mutant virus, the presence of the intended mutation was confirmed by sequencing RT-PCR product. WSN virus encoding FLAG-tagged PB2 [51] was used for infections for mass spectrometric analysis and cRNA stabilization assays.

Transfections were performed using either TransIT 2020 (Mirus MIR5400) or PEI MAX40 (Polysciences 24765–1) following the manufacturers' recommendation.

### Antibodies

FLAG-PB2 purifications for mass spectrometry were performed using M2 antibody (Sigma F1804) and captured using protein A dynabeads (Invitrogen 10002). Other FLAG immuno-precipitatons were performed with M2 Affinity Gel (Sigma A2220). The following antibodies were used for western blot analysis: α-PB1 [48], α-PB2 [48], α-PA (Genetex 125933), HA-HRP (3F10, Sigma 12013819001), M2-HRP (Sigma 8592). The mouse α-NP monoclonal antibody H16-L10-4R5 (Bioxcell BE0159) [52] was used for both immunoprecipitation and western blot analysis of NP.

### Effects of PKCδ on polymerase function

HEK 293T cells were transfected to express PB1, FLAG-tagged PB2, PA and vNP77 [33]. Constitutively active HA-PKCδ was co-transfected where indicated. PB2 was immuno-purified, treated with calf intestinal phosphatase or mock treated, and analyzed by western blotting. Viral RNAs were detected and quantified by primer extension. Western blotting of whole cell extract was used to test equivalent expression of viral proteins in all conditions.

### Synchronized single-cycle infections and multicycle replication assays

Viral infections with A549 cells were performed in virus growth medium (DMEM, 0.2% BSA, 25mM HEPES, 0.25 μg/mL TPCK-trypsin). Viral infections with 293T cells were performed in OptiMEM (Invitrogen) containing 2% FBS. For synchronized infections, cell monolayers were washed twice with ice-chilled PBS, incubated with inoculum for 1 hr at 4˚ C, followed by removal of inoculum and addition of pre-warmed fresh VGM (37˚ C) [53]. Cells were maintained at 37˚ C for the duration of the infection. Multicycle replication assays were performed in A549 cells by inoculating cells in triplicate at an MOI 0.001. Virus was sampled at the indicated time points and titers were determined by plaque assay on MDCK cells.

Gene expression was measured during an asynchronous infection by inoculating A549 cells at an MOI of 0.05 for 8 h. Infections were terminated and total RNA was extracted from cells using TRIzol (Invitrogen). 250 ng of total RNA was subject to poly-dT primed reverse transcription using MMLV-RT. Resulting cDNA was used for qPCR to detect GAPDH and NP mRNA with the iTaq Universal SYBR Green Supermix (Bio-Rad 1725121). Fold changes in NP mRNA were calculated using the $\Delta\Delta C_T$ method.

## Polymerase activity assays

HEK 293T cells were transfected with plasmids expressing NP, PB2, FLAG-tagged PB1, FLAG-tagged PA, and segment 6 vRNA (NA). Total RNA was extracted using TRIzol 24 hr after transfection.

## Primer extension analysis

RNA was subject to primer extension analysis for genomic (vRNA), complementary (cRNA), and messenger (mRNA) corresponding to segment 6. Primer extension analysis was performed as previously described for full-length and short (NP77) templates [54,55]. Briefly, RNA and the appropriate radiolabeled primers were boiled for 2 min and snap-chilled on ice. Samples were pre-heated to 42˚ C and pre-heated reaction mixture was added for a final reaction containing 50 mM Tris-HCl (pH 8.3), 75 mM KCl, 3 mM $MgCl_2$, 5 mM DTT, 40 units RNAsin+ (Promega N2611), and house-made MMLV-RT [55]. Samples were incubated for 1 hr at 42˚ C. Reactions were terminated with an equal volume of 2x RNA loading dye (47.5% formamide, 0.01% SDS, 0.5 mM EDTA containing bromophenol blue and xylene cyanol), boiled for 2 mins and snap-chilled on ice prior to resolving on 6% (full-length templates) or 12% (short templates) denaturing polyacrylamide gels containing 0.5X TBE and 7M Urea. Gels were fixed (40% methanol, 10% acetic acid, 5% glycerol) for 30 mins, dried, quantified by phosphorimaging, and analyzed using Image Studio software (Licor).

## Preparation of samples for mass spectrometry

Mass spectrometry was performed on both transfected and infected cells. HEK 293T cells (3 x 100mm dishes containing approximately $6x10^6$ cells) were transfected to express PB1, FLAG-tagged PB2, PA and vNP77 [33]. 24 hr later, cells were lysed in RIPA Buffer (150mM NaCl, 50mM Tris pH 7.5, 0.5% w/v sodium deoxycholate, 0.1% w/v SDS, 1% w/v Ipegal 630, 2mM EDTA) in the presence of protease inhibitors and phosphatase inhibitors for 20 mins at 4˚C. Lysates were sonicated and then cleared via centrifugation at 4˚ C. FLAG-tagged PB2 was immunoprecipated using M2 antibody overnight at 4˚ C. Immunocomplexes were captured using protein A dynabeads (Invitrogen 10002D) for 2 hr. Immunoprecipitations were washed twice with RIPA buffer and 4 times with NTE (100mM NaCl, 10mM Tris pH 7.5, 1mM EDTA). Protein was eluted with 8M urea in NTE. Samples were frozen at -80˚ C prior to mass spectrometric analysis.

For samples prepared from virally-infected cells, A549 cells were synchronously infected with WSN at an MOI of 5. Cells were collected at 2.5 hpi ($30x10^6$ cells), 5 hpi ($18x10^6$ cells), 7.5 hpi ($18x10^6$ cells), and 10 hpi ($12x10^6$ cells). Cell numbers were adjusted in an attempt to account for lower amounts of viral proteins early during infection. Cells were scraped into ice-chilled PBS and collected by centrifugation. NP immunoprecipitations were performed as above using α-NP monoclonal antibody H16-L10-4R5. Samples were also prepared using the same approach in separate experiments where A549 cells were infected with WSN at an MOI of 0.1 for 24 hr or an MOI of 5 for 6hr. Cell pellets and immunoprecipitations were frozen at -80˚ C prior to mass spectrometric analysis.

## Mass spectrometry

**Sample preparation for Nano-LC-MS/MS.** Infected cells were lysed in 6M guanidine-HCl for 10 min at 100˚C. Protein was precipitated by addition of methanol to a final concentration of 90% and pelleted by centrifugation at 12,000 x G for 10 min. The supernatant was discarded and pellet was resuspended in 8M urea, 50 mM Tris (pH 8.0), 10 mM tris(2-carboxyethyl)phosphine) (TCEP) and 40 mM chloroacetamide and rocked at room temperature for 30 min to reduce and alkylate cysteines. The sample was diluted to a urea concentration of less than 1.5 M with 50 mM Tris (pH 8.0) before adding protease grade trypsin (Promega) at an enzyme:protein ratio of 1:50 (mg:mg). The samples were rocked overnight at room temperature during the digestion. 10% trifluoroacetic acid was added to the solution to bring the pH of the sample less than 2 before desalting and peptide isolation using Strata-X reverse phase resin (Phenomenex). Sample were dried under reduced pressure, resuspended in 0.2% formic acid, and quantified by Pierce Quantiative Colorimetric Peptide Assay (Thermo Fisher Scientific). Phosphopeptides were enriched for each sample from 2 mg tryptic peptides using immobilized metal affinity chromatography (Ti-IMAC MagResyn, ReSyn Biosciences).

**Nano-LC-MS/MS data acquisition.** Each sample was analyzed using an Q-LTQ-OT tribrid mass spectrometer (Orbitrap Fusion Lumos) during a 90 min nano-liquid chromatography using a Dionex UltiMate 3000 RSLCnano system (Thermo Fisher Scientific). MS parameters differed for the analysis of phosphopeptides enriched and unenriched sample. For unenriched sample, MS1 survey scans were acquired in the Orbitrap (Resolution– 240K, AGC Target– $1x10^6$, Scan Range– 300–1,350 Da, Maximum Injection Time– 100 ms). MS2 spectra of observed precursors were acquired in the ion trap (Resolution–Rapid, AGC target $4x10^4$, Scan Range– 200–1,200, Maximum Injection Time– 18 ms) following quadrupole isolation (0.7 Da) and higher energy collisional dissociation (25% NCE). For phosphopeptides enriched samples, MS1 survey scans were acquired in the Orbitrap (Resolution– 60K, AGC Target– $1x10^6$, Scan Range– 300–1,350 Da, Maximum Injection Time– 50 ms). Observed precursors were also analyzed in the Orbitrap (Resolution– 15K, AGC Target– $5x10^4$, Scan Range– 150–1,500, Maximum Injection Time– 50 ms) following quadrupole isolation (1.6 Da) and higher energy collisional dissociation (25% NCE). Monoisotopic precursor isolation and a dynamic exclusion of 15 s were enabled for both methods.

**Data analysis.** Thermo RAW data files were searched using MaxQuant (version 1.5.3.51) with the Andromeda search algorithm against a concatenated target-decoy database of human and influenza proteins using default search tolerances [56,57]. Specified search parameters included the fixed modification of carbamidomethylation at cysteine residues and variable modification for methinine oxidation. Phosphorylation of serine, threonine, and tyrosine were specified as variable modifications for phosphopeptide enriched data. Label free quantitation and intensity based absolute quantitation were enabled [58,59]. Raw spectra files are available upon request.

## Amino acid conservation analysis

Amino acid conservation analysis was performed using the Analyze Sequence Variation (SNP) tool available at the NIAID Influenza Research Database (fludb.org) [60]. The percentage of sequences encoding serine, threonine, and tyrosine was calculated from precompiled sets of influenza virus sequences corresponding to human H1N1, H3N2, and H5N1 isolates.

## Surface accessibility analysis

Solvent accessibility of identified phosho-residues was determined using GETAREA software with the following structures: 4WSB (vRNA bound), 6KUJ (cRNA bound), 6QXE (cRNA

bound dimer), and 6T0V (transcription elongation) [21,35,36,44,61]. Calculated surface accessible surface area (SASA) of each residue < 20% or > 50% of the average SASA of the corresponding Gly-X-Gly peptide were scored as buried or accessible, respectively.

## Polymerase formation assays

Polymerase assembly was measured as before [62]. FLAG-tagged PB1, PB2, and PA were expressed in transfected HEK 293T cells for 48 hr. Cells were lysed in co-IP buffer (50mM Tris pH 7.4, 150mM NaCl, 0.5% Igepal CA-630) in the presence of protease inhibitors for 20mins at 4˚ C. Lysates were clarified by centrifugation and pre-cleared with protein-A agarose (Santa Cruz Biotech sc-2001) for 1hr. Lysates were then transferred to a new microcentrifuge tube and BSA was added to a final concentration of 5 mg/ml. FLAG-PB1 was immunoprecipitated overnight with M2-agarose. Immunoprecipitations were washed twice with co-IP buffer containing 5 mg/mL BSA and 500mM NaCl and twice with co-IP buffer. Bound proteins were eluted by boiling in Laemmli buffer. Samples were then assayed via western blot analysis for presence of PB1, PB2, and PA.

## RNP reconstitution assays

NP, PB2, FLAG-tagged PB1, FLAG-tagged PA, and segment 6 vRNA (NA) were expressed in transfected HEK 293T cells for 48 hr, following prior approaches [54]. Cells were lysed in co-IP buffer in the presence of protease inhibitors. Lysates were clarified by centrifugation, pre-cleared protein A agarose (Santa Cruz Biotech sc-2001) for 1 hr, and transferred to a new tube where BSA was added to a final concentration of 5 mg/mL. NP was immunoprecipitated overnight with 3 μg anti-NP antibody. Immunocomplexes were captured using protein A agarose (Sigma P2545) for 1 hr, washed twice with co-IP buffer containing 5 mg/mL BSA and 500 mM NaCl, and twice with co-IP buffer. Bound proteins were eluted by boiling in Laemmli buffer. Samples were then assayed via western blot analysis for presence of NP, PB1, and PA.

## cRNA stabilization assay

cRNA stabilization was measured as previously described [16,63]. Briefly, HEK 293T cells were transfected to express the viral polymerase with the indicated PB1 subunit and an oligomerization deficient NP ($NP_{E339A}$). 24 hr post-transfection, cells were treated with actinomycin D (5 μg/mL) (Sigma A1410) for 30 mins prior to asynchronous infection with WSN in the presence of actinomycin D. Cells were harvested 6 hpi. Total RNA was extracted using TRIzol and used in primer extension analysis.

## RNA immunoprecipitation vRNA binding assay

RNPs with FLAG-tagged PB1 were reconstituted in HEK 293T cells as above. Cells were lysed 48 hr post-transfection in co-IP buffer supplemented with both protease inhibitors and RNA-sin (Promega N2515, 100 units/mL). Lysates were processed and immunoprecipitations were performed as described above for the polymerase formation assay. Protein from 10% of the immunoprecipitate was eluted by boiling in Laemmli buffer and assayed via western blot. RNA from 90% of the immunoprecipitate was extracted using TRIzol and analyzed by primer extension.

## Statistics

Data represent at least 2–3 independent biological replicates. Technical replicates are indicated for each figure. Quantitative data are shown as mean ± standard deviation for one biological

replicate or the mean of means ± standard error of measurement for multiple biological replicates. Single pair-wise comparisons were analyzed by two-tailed Student's t-test. Multiple comparisons were performed by a one-way ANOVA followed by Dunnett's *post hoc* analysis of pair-wise comparisons to WT. P<0.05 was considered significant. Statistic were calculated in Prism 8.

## Supporting information

**S1 Fig. Quantification of replicate primer extension assays.** Three independent primer extension assays were quantified with each RNA species normalized to WT within an experiment. Data are presented at mean ± sd. * < 0.05, ** < 0.01, *** < 0.001 = P for one-way ANOVA with Dunnett's *post hoc* compared to WT.
(TIF)

**S2 Fig. Quantification of replicate cRNA stabilization assays.** vRNA levels were quantified from three independent stabilization assays and normalized to WT. Data are presented at mean ± sd. There was no significant difference when analyzed by a one-way ANOVA.
(TIF)

**S3 Fig. The position of PB1 S673 throughout the influenza virus polymerase transcription cycle.** Structures of the polymerase at different stages of the catalytic cycle. Portions of PB1 are shown in purple with the motif C residues D445/D446 in the catalytic site in orange. 5' vRNA is light blue, 3' vRNA is magenta, and transcription product is dark blue. The PB2-N1 domain is modeled in orange. **A.** PB1 S673 (yellow) and PA H510 (cyan) flank the incoming 3' vRNA template in a pre-initiation state (PDB 6T0N). **B-D.** Residues upstream of the PB1 α20 helix remodel to create the template exit channel used during transcription elongation (PDB: 6T0V). PB1 S673 is repositioned at the floor of the exit channel and remains part of the exit channel during later stages of transcription including polyadenylation (PDB: 6T0S) and pre-termination (PDB: 6SZU). Number and residue assignments are based on WSN sequences.
(TIF)

**S1 Table. Identification of PB1 phosphorylation sites by mass spectrometry.** Summary Data presents a composite of all sites identified in each experimental condition. See Fig 2A for experimental design. Other tabs represent underlying data for phospho-site identification in each experiment. Note that not all sites are found in all conditions, or at all time points.
(XLSX)

**S2 Table. Surface accessibility of phosphorylated PB1 residues.**
(XLSX)

**S3 Table. Statistical analysis of multicycle replication.** Adjusted P value for comparison between WT and PB1 mutants during multi-cycle replication in Fig 3B using a one-way ANOVA with Dunnett's *post hoc* correction.
(XLSX)

## Acknowledgments

We thank members of the Mehle and Coon lab for their constructive input.

## Author Contributions

**Conceptualization:** Anthony R. Dawson, Arindam Mondal, Andrew Mehle.

**Data curation:** Anthony R. Dawson, Gary M. Wilson, Elyse C. Freiberger.

**Formal analysis:** Anthony R. Dawson, Gary M. Wilson, Elyse C. Freiberger, Joshua J. Coon, Andrew Mehle.

**Funding acquisition:** Anthony R. Dawson, Gary M. Wilson, Joshua J. Coon, Andrew Mehle.

**Investigation:** Anthony R. Dawson, Gary M. Wilson, Elyse C. Freiberger, Arindam Mondal.

**Methodology:** Anthony R. Dawson, Gary M. Wilson, Elyse C. Freiberger, Arindam Mondal.

**Project administration:** Joshua J. Coon, Andrew Mehle.

**Resources:** Anthony R. Dawson.

**Software:** Gary M. Wilson, Elyse C. Freiberger.

**Supervision:** Joshua J. Coon, Andrew Mehle.

**Validation:** Anthony R. Dawson, Gary M. Wilson.

**Visualization:** Anthony R. Dawson, Andrew Mehle.

**Writing – original draft:** Anthony R. Dawson, Gary M. Wilson, Joshua J. Coon, Andrew Mehle.

**Writing – review & editing:** Anthony R. Dawson, Gary M. Wilson, Joshua J. Coon, Andrew Mehle.

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
