## [Decision Letter · Decision Letter 0]

5 Jun 2020

Dear Dr Mehle,

Thank you very much for submitting your manuscript "Phosphorylation controls RNA binding and transcription by the influenza virus polymerase" for consideration at PLOS Pathogens. As with all papers reviewed by the journal, your manuscript was reviewed by members of the editorial board and by several independent reviewers. The reviewers appreciated the attention to an important topic. Based on the reviews, we are likely to accept this manuscript for publication, providing that you modify the manuscript according to the review recommendations.

Sincerely,

Shin-Ru Shih

Associate Editor

PLOS Pathogens

Andrew Pekosz

Section Editor

PLOS Pathogens

Kasturi Haldar

Editor-in-Chief

PLOS Pathogens

orcid.org/0000-0001-5065-158X

Michael Malim

Editor-in-Chief

PLOS Pathogens

orcid.org/0000-0002-7699-2064

Reviewer Comments (if any, and for reference):

Reviewer's Responses to Questions

**Part I - Summary**

Reviewer #1: The authors have adequately addressed previously raised concerns.

Reviewer #2: In this paper Dawson et al. identify multiple phosphorylation sites in the PB1 subunit of influenza A virus polymerase using mass spectrometry. Using RNP reconstitution and primer extension assays the authors demonstrate mechanistic effects of PB1 phosphorylation on the ability of viral polymerase to transcribe and replicate viral RNA. Finally, using viral reverse genetics, the authors also show that some of the discovered phosphorylation sites are required for virus replication in tissue culture.

Overall, the presented research supports the authors’ conclusion and presents findings about the modulation of viral genome replication by viral polymerase phosphorylation that will be of interest to the influenza research community.

Reviewer #3: The authors use mass spectrometry to identify phosphorylation sites of influenza virus polymerase, either transfected together with active PKCdelta or in infected cells. Three out of 13 sites in PB1 are common to both kinds of experiment. They focus on five sites and by analysing phospho-mimetic and phospho-ablative mutants, in recombinant viruses or transfections, show that the phosphorylation status modulates polymerase activity in diverse ways. Such information is new and beneficial, as a starting point, for improving understanding of polymerase regulation during viral infection. However, in view of the vast amount of structural information now available on influenza polymerase in different functional states, especially during transcription, the discussion of the mechanism of these phosphorylation effects is disappointing and should be further developed. Structural studies show that the polymerase is incredibly flexible, locally and globally, especially in the apo-state, and has multiple interchanging functional conformations. This undoubtedly affects accessibility of certain residues to kinases and phosphatases. Furthermore, many residues have multiple roles. For example, PB1 S673 helps stabilise a particular conformation of the 3’ end in the promoter, but during elongation it is in the outgoing template channel. Similarly, PB1 T570, which the authors unfortunately do not study in detail, is in the newly described secondary 3’ end binding site and its phosphorylation could inhibit RNA binding there. Residues S712 and S720 are both in the interface between the PB1 core and the PA endonuclease domain, but this is not mentioned.

**Part II – Major Issues: Key Experiments Required for Acceptance**

Reviewer #1: (No Response)

Reviewer #2: No major comments. Authors have appropriately addressed the previous major from the reviewers and updated their manuscript accordingly.

Reviewer #3: 1. The authors make poor use of available structural information to help the reader think about possible effects of the observed phosphorylation sites. Rather than just categorising as catalytic or template entry, they should provide a table with a short description of the structural position of the residue and its possible functional role (e.g. T570: in secondary 3’ end binding site, S673: stabilises kink in 3’ end of promoter/during elongation in template exit channel, S712, S720: in PB1-endonculease interface, S384, T385: close to tip of 5’ hook, Y253: in active site cavity, S478: in NTP entrance channel close to incoming NTP triphosphate binding site etc).

2. Figure 2B should be improved to show all identified sites in the global view and the detailed part on top of the global view should be removed. There could be two enlarged view, one of the promoter bound form, one of the transcription elongation form.

3. The authors should comment on the accessibility of the individual residues early on (and not only in the discussion), to give the reader an impression how reversible individual modifications might be. The authors should not just say ‘residues are not immediately on the surface’ but calculate solvent accessibility in different polymerase states (compare for example the transcriptionally ready 4WSB and the cRNA bound 6KUJ, or the recent polymerase dimers e.g. 6QX8). The authors might also emphasise that some of the sites (especially those around RNA binding regions) are probably more accessible when the polymerase is RNA free.

4. The authors find T223D eliminates vRNA binding and cRNA stabilisation, yet, according to structures, it is not in close contact with viral RNA. On the other hand S673 directly binds the promoter 3’ end and thus S673D would be more expected to effect vRNA binding. The authors should perform the same vRNA and cRNA binding studies on the S673D mutant and try to explain this counter-intuitive phenotypes of these two mutations.

5. Lines 275-276. The authors suggest there is evidence that the 3’ end of the template is differently located dependent on whether it is undergoing transcription or replication initiation. Specifically, from its structurally observed position in the active site, the 3’ end appears to need to backtrack by one nucleotide to allow terminal initiation of vRNA to cRNA replication, although using the same reasoning (based on the pathway of the template going into the active site being always the same), would apparently not need to do so to allow internal initiation of cRNA to vRNA initiation. The observation that S673D specifically affects transcription but not either sense of replication suggests that, given its key structural role in stabilising the G9-C8 kink (which would seemingly be totally disrupted if S673 were phosphorylated) in the observed promoter structure, the template pathway may not be the same and indeed the whole promoter might be different during replication. This is a very significant observation that merits more careful discussion (and a figure showing the dual roles of S673 in promoter binding and in elongation). For instance, another nearby residue, intimately involved in binding the promoter, PA H510, upon mutation also specifically affects transcription (J Virol. 2002;76(18):8989‐9001. doi:10.1128/jvi.76.18.8989-9001.2002).

**Part III – Minor Issues: Editorial and Data Presentation Modifications**

Reviewer #1: (No Response)

Reviewer #2: No minor comments. Authors have appropriately addressed the previous minor from the reviewers and updated their manuscript accordingly.

Reviewer #3: 1. Remind the reader whether PKCdelta is active in the nucleus or cytoplasm or both.

2. Is it possible to quantify from the mass-spec analysis the fraction of polymerases that have a particular phosphorylation?

3. Expand the phospho-residue identity search to bat influenza A and influenza B. Sites really important for modulation of intrinsic functions are likely to be fully conserved. For instance S384, whose A/D mutations are pretty much like wild-type, is already an aspartate in bat influenza polymerase.

4. On lines 268-284 the authors describe the effects of S673 phosphorylation. Indeed, the S673 phosphorylation may influence the behaviour of template RNA in this region. However speculation in lines 281 about charge neutralisation is not supported by the structure organisation of the region.

5. The authors speculate that some of the inaccessible sites might be modified already during polymerase translation and are static/irreversible (lane 307). The T223 and S478 are actually an example of inaccessible residues. More over the S478D is completely inactive. How do the authors explain what happens when such a polymerase is incorporated into a newly made RNP (lane 289) thus making it incapable of any further transcription and replication. Such a contradiction should be explained in comparison to positive effect of activation of RNPs in trans (lane 293).

6. Line 302-304 unsupported speculation whose relevance is dubious.

7. Line 306. This is a very static view of the polymerase, whereas it is actually extremely flexible.

PLOS authors have the option to publish the peer review history of their article (what does this mean?). If published, this will include your full peer review and any attached files.

Reviewer #1: No

Reviewer #2: No

Reviewer #3: No
---

## [Editor Report · Decision Letter 1]

25 Jul 2020

Dear Dr Mehle,

We are pleased to inform you that your manuscript 'Phosphorylation controls RNA binding and transcription by the influenza virus polymerase' has been provisionally accepted for publication in PLOS Pathogens.

Best regards,

Shin-Ru Shih

Associate Editor

PLOS Pathogens

Andrew Pekosz

Section Editor

PLOS Pathogens

Kasturi Haldar

Editor-in-Chief

PLOS Pathogens

orcid.org/0000-0001-5065-158X

Michael Malim

Editor-in-Chief

PLOS Pathogens

orcid.org/0000-0002-7699-2064
---

## [Editor Report · Acceptance letter]

28 Aug 2020

Dear Dr Mehle,

We are delighted to inform you that your manuscript, "Phosphorylation controls RNA binding and transcription by the influenza virus polymerase," has been formally accepted for publication in PLOS Pathogens.

Best regards,

Kasturi Haldar

Editor-in-Chief

PLOS Pathogens

orcid.org/0000-0001-5065-158X

Michael Malim

Editor-in-Chief

PLOS Pathogens

orcid.org/0000-0002-7699-2064